# Exploring the Role of Personal Demands in the Health-Impairment Process of the Job Demands-Resources Model: A Study among Master Students

**DOI:** 10.3390/ijerph18020632

**Published:** 2021-01-13

**Authors:** Marijntje E. L. Zeijen, Veerle Brenninkmeijer, Maria C. W. Peeters, Nicole J. J. M. Mastenbroek

**Affiliations:** 1Department of Population Health Sciences, Faculty of Veterinary Medicine, Utrecht University, 3508 TD Utrecht, The Netherlands; n.j.j.m.mastenbroek@uu.nl; 2Department of Social and Organizational Psychology, Utrecht University, 3508 TC Utrecht, The Netherlands; v.brenninkmeijer@uu.nl (V.B.); m.peeters@uu.nl (M.C.W.P.); 3Human Performance Management Group, Eindhoven University of Technology, 5600 MB Eindhoven, The Netherlands

**Keywords:** personal demands, study demands, burnout, JD-R model, exhaustion

## Abstract

Research shows that students experience substantial levels of burnout during their studies. This study explores the role of personal demands on students’ well-being. After providing a conceptualization of personal demands, we examined the role of personal demands in the Job Demands-Resources (JD-R) model. Based on the Transactional Model of Stress, we hypothesized that students with high personal demands experience more burnout symptoms because they perceive more elements in their study as demanding (i.e., mediation hypothesis). At the same time, we hypothesized that the associations between study demands and burnout might be stronger for students with high versus low personal demands (i.e., moderation hypothesis). In order to test both hypotheses, we collected data from 578 master students. The data were analyzed with latent moderation and mediation analyses in Mplus. The results showed that students’ personal demands predicted burnout symptoms via the perception of study demands. Personal demands did not moderate the relationship between study demands and burnout. The findings of the present study expand the JD-R model by indicating that personal demands relate to burnout symptoms via the perception of study demands. Theoretical and practical implications are discussed.

## 1. Introduction

Feeling exhausted, losing control over one’s emotions, and mentally distancing oneself from one’s studies are nowadays common experiences among students. Empirical studies on student well-being indicate that students in higher education experience substantial levels of burnout symptoms during their studies [1,2,3]. These burnout symptoms seem to be partly caused by the level of study demands that students experience [4,5,6,7]. However, students’ well-being may also be influenced by personal aspects that impede effective management of one’s study environment. Workaholism, for instance, has been found to be a relevant demanding personal factor for employee well-being [8]. As it is not yet clear what the role of demanding personal aspects are in relation to student burnout, we will examine this issue in the present study.

Theoretically, the Job Demands-Resources model (JD-R model) [9] underscores the role of demanding aspects of individuals’ work (or study) environment as a risk factor for burnout. Applying the JD-R model to a student’s situation, any study can be considered as consisting of a constellation of study demands (e.g., workload, emotional demands, and time pressure) and study resources (e.g., social support, feedback, and autonomy), which respectively deplete or boost students’ energy. Specifically, the process through which study demands impair students’ energy is labeled as the health impairment process, while the process through which study resources boost students’ motivation is labeled as the motivational process. In addition to job or study related characteristics, the JD-R model includes personal resources, referring to all aspects of the self that generally link to resilience and reflect an enhanced self-perceived ability to successfully influence one’s environment [10,11]. Personal resources have been found to positively predict job resources [9] and to strengthen the positive relationship between job resources and work-related well-being [9,12,13].

So far, research has paid little attention to personal aspects that hinder individuals to successfully influence their work or study environment (for notable exceptions, see: [8,14,15]). These studies point to the relevance of workaholism [8,15], external locus of control [14], and perfectionism [15] for the perception of job demands and employee burnout. It is surprising that hindering personal aspects have gained little attention in research, as the investigation of these aspects, often referred to as dysfunctional beliefs, is considered to be essential for unraveling the etiology of psychological problems such as burnout [16,17,18]. Therefore, in order to gain insight into why students experience elevated levels of burnout symptoms during their studies, we focus in the present study on how hindering aspects on the personal level (i.e., students’ personal demands) relate to the perception of their study environment and to the experience of burnout symptoms. As such, we examined how the concept of personal demands fits into the health impairment process of the JD-R model (see the research model in Figure 1).

In doing so, this research contributes to the literature in three significant ways. First of all, by elaborating on the role of relevant personal demands for students’ well-being, this research enhances insight into the etiology of (student) burnout (e.g., [5,6,19]). Whereas the JD-R model predicts how students’ personal resources may help them to be successful while staying healthy, this study offers novel insights into how students’ personal demands may deplete energy and increase the risk of exhaustion and burnout. Secondly, by providing a theoretically grounded conceptualization of personal demands and empirically testing its evident place within the JD-R model, this research introduces personal demands into the JD-R model and, as such, follows up on a call from [9] to include personal demands into the JD-R model. Finally, this research aimed to contribute to the toolkit of professionals responsible for the prevention, treatment, and coaching of students with burnout complaints. By investigating the role of students’ personal demands in relation to the perception of their study environment and burnout, this research offers insights for practitioners for targeting specific personal demands in order to reduce burnout among students.

### 1.1. Conceptualizing Personal Demands

Barbier, Hansez, Chmiel, and Demerouti [20] were among the first to explicitly define personal demands, namely as “the requirements that individuals set for their own performance and behavior that force them to invest effort in their work and are therefore associated with physical and psychological costs” [20] (p. 751). Personal demands give rise to experiences of internal pressure [20,21] and depend on the values and needs of individuals who translate these needs into objectives, motivation, and effort to achieve these objectives [22]. Although we believe that this definition contains several important elements of the concept of personal demands, we introduce three refinements in order to create a better fit with the JD-R model [9].

First, while we agree with the proposed mechanism that setting requirements for oneself forces people to invest effort [20], we propose that this may not be the only mechanism that forces individuals to invest effort. Previous studies have revealed that a broad array of self-related aspects cause individuals’ energy to be consumed and not replenished. Examples of such mechanisms are irrational beliefs, such as awfulizing, and an irrational need for control [23]. Therefore, in order to capture the full array of potential personal mechanisms in the definition of personal demands, we propose to consider all possible aspects of the self that may force people to invest effort. An advantage of adopting the definition to refer to all aspects of the self is that we streamline the conceptualization of the personal demands with the conceptualization of personal resources in the JD-R model, which also refers to all aspects of the self [11].

Another refinement that we would like to add to the proposed definition of personal demands refers to the term disproportionate when referring to personal investment. By referring to the disproportionality of effort investment, compared to the outcomes that are pursued, we emphasize the maladaptive nature of personal demands and aim to distinguish the concept of personal demands more distinctly from personal resources in this research. After all, investing personal resources in order to successfully influence one’s environment asks for the expenditure of psychological or physical energy as well [24]. By definition, personal resources manifest successful outcomes, and the effort that is invested in influencing one’s work or study environment can be considered as adaptive and proportionate with regard to the outcomes that are pursued. Therefore, by adding “disproportionate investment” to the definition of personal demands, we underline the maladaptive nature of personal demands.

The final refinement that we propose is to add “aspects of the self that hamper individuals to successfully cope with the demands in their environment”. This means that rather than solely capturing personal aspects that force individuals to invest extra effort, we find it relevant to pay also attention to personal aspects that withhold individuals from properly responding to demanding aspects in their environment. According to Ohue et al. [18], personal aspects such as problem avoidance, dependency, and perceived helplessness form specific demanding cognitive schemas that fuel negative thoughts and emotions, may eventually lead to burnout. In line with these theoretical and empirical statements, we argue that personal aspects that restrict individuals to successfully cope with their environment also belong to the concept of personal demands.

Taken together, in the present study, we refer to personal demands as all aspects of the self that force individuals to invest disproportionate effort in their work and/or hamper them to successfully cope with their environment and are therefore associated with psychological and/or physical costs.

### 1.2. Personal Demands in the Health-Impairment Process

Following the definition of personal demands that we proposed above, we assume that personal demands, such as perfectionism, dependency, or an irrational need for control, constitute a relevant factor for the health impairment process within the JD-R model. As students with high personal demands may either force themselves to invest a disproportionate amount of effort into their studies or may be hampered to successfully cope with the demands in their study, they may be more prone to experiencing stress and exhaustion. Nonetheless, it has yet to be unraveled how personal demands exactly influence the health impairment process.

In the present study, we investigated how personal demands relate to the experience of study demands and burnout symptoms among students. In particular, we build our theoretical reasoning on the Transactional Model of Stress [17], since the appraisal of situations and the extent to which individuals experience a situation as demanding and stressful lies at the core of this model. A central component of the Transactional Model of Stress is the subjective appraisal of any given situation. This means that, rather than the objective aspects of a situation, individuals’ personal and subjective appraisal determines to what extent a situation is experienced as demanding and stressful [17]. The authors further argue that the appraisal process follows two paths: Primary appraisal, in which is appraised how demanding the situation in itself may be, and secondary appraisal, in which is appraised whether one is able to cope with the situation.

Additionally, we build further upon empirical studies that have highlighted the relevance of work-related irrational cognitions for the way in which people experience and react to work situations [23,25]. Based on the findings of these studies, it can be expected that personal demands will influence the extent to which study situations are perceived as demanding. For instance, previous findings have indicated that employees with strong irrational beliefs about their performance (e.g., “I must do my work flawlessly”) approach their work in a more obsessive and compulsive manner (e.g., “I overly commit myself by biting off more than I can chew”) [23]. This implies that employees who, for example, demand a flawless performance from themselves are more likely to perceive their work as highly demanding. In a similar vein, a previous study on the relationship between the appraisal of daily hassles and headaches among students suggested that perfectionistic students are more likely to generate their own stress as they tend to perceive more situations as hassles [26].

Referring back to the primary and secondary appraisal pathways from the Transactional Model of Stress [17], we theorize that students’ personal demands may influence the process of primary appraisal of their study environment. Hence, we assume that personal demands press students to perceive any study situation as being more demanding. Taken together and building further on the existing evidence that study demands positively relate to exhaustion and burnout [4,5,6,7,19], we formulate the following hypothesis:

**Hypothesis** **1** **(H1).**
*Personal demands are positively associated with burnout via the perception of high study demands (i.e., mediation hypothesis).*


In addition, students’ personal demands may influence the process of secondary appraisal of their study environment, during which is assessed to what extent one is capable of dealing with in a given situation [17]. Following the definition of personal demands that we proposed above, we assume that students’ personal demands may hamper students to successfully handle challenging study situations. For instance, students with an irrationally high need for control may perceive a difficult assignment (i.e., a high study demand) as particularly stressful because they believe that they are not able to cope with the unforeseen, challenging aspects of the assignment. Individuals with a high need for control find it difficult to cope with situations that are unpredictable [26], while, at the same time, they are more prone to experience exhaustion and burnout [27].

So far, studies testing moderation effects of personal demands on the relationship between situational demands and ill-being seem to be scarce. A few empirical studies exist that reveal the negative impact of personal demands on the situational demands and ill-being relationship. For instance, a previous study showed that salespeople with an external locus of control (who believe that they are unable to influence their surroundings) experience more stress when confronted with excessive work demands as compared to salespeople with an internal locus of control [28]. Another study showed that individuals with an external locus of control experience more somatic health problems when confronted with role conflict, as compared to individuals with an internal locus of control [29]. In line with these empirical findings and the secondary appraisal process of the Transactional Model of Stress [17], we presently argue that personal demands make it more difficult for students to successfully handle challenging study requirements and to feel confident about their capability for successful enactment, which is likely to relate to stress and exhaustion. Hence, we also contended the following:

**Hypothesis** **2** **(H2).**
*Personal demands moderate the positive relationship between study demands and burnout, in the sense that the relationship between study demands and burnout is stronger when personal demands are high (vs. low).*


## 2. Materials and Methods

### 2.1. Participants

As part of a research project on the well-being of master students, exploratory group interviews were conducted among 12 veterinary medicine students, five humanities students, four law students, and six biomedical medicine students participated. Subsequently, survey data were collected in May and June 2018. In total, 1955 master students from four different faculties of one university in the Netherlands were invited to participate in the survey study. A total of 658 students agreed to participate. Of these 658 participants, 79 participants were omitted from the sample as they had missing data on all variables of interest, and one participant was dropped due to double participation. In total, 578 students were included in the study (response rate of 30%). Participants were master students from three veterinary medicine master programs (*n* = 271), from ten Law master programs (*n* = 127), from ten biomedical sciences master programs (*n* = 145), and from five humanities master programs (*n* = 35). The male/female ratio for humanities (29/71%) and veterinary medicine (14/86%) were representative for their faculty, whereas, for biomedical sciences (23/76%) and law (31/69%), relatively more women participated. Students were on average M = 24.37 (SD = 2.9) years old. There were no significant age differences between participating faculties (*p* = 0.157).

### 2.2. Procedure

The composition of the questionnaire was realized in two phases. In the first exploratory phase, semi-structured group interviews were conducted in order to uncover which personal demands and study demands were most relevant for master students from faculties that participated in this study. As the research project initially started within the faculty of veterinary medicine, two group interviews with veterinary medicine master students were conducted. For faculties who joined the project afterward (biomedical sciences, law, and humanities), one group interview per faculty was deemed sufficient. Students were asked what aspects of their studies drained their energy and what personal traits, beliefs, or characteristics hindered them in their study. Invitations to participate in the group interviews were communicated to all students of the participating faculties by the student representation of these faculties. No incentives for participation in group interviews were offered. The group interviews were conducted by members of the research team. These researchers were not related in any way to the participants of the group interviews. Results of the group interviews were analyzed by the members of the research team with directed content analysis [30]. Based on the results of the first exploratory phase, the questionnaire for the survey study was developed. The questionnaire was piloted among a limited number of staff members and students within the faculty of Veterinary Medicine. Subsequently, master students from all participating faculties were invited to participate via email in the survey study. All participants were informed that participation was voluntary and anonymous and that they could quit at any moment. In reward for their participation, participants were given the option to participate in a lottery in which students had a chance to win a wellness gift card (handed out five times worth €50). Participants were also given the option to receive individual feedback on their burnout scores. All students were reminded twice to participate in the study, except for biomedical master students, who were reminded once due to the faculty’s emailing policy.

### 2.3. Ethical Approval

Prior to executing the study, ethical approval was gained from the Ethical Review Board of the Dutch Association for Medical Education (i.e., Nederlandse Vereniging voor Medisch Onderwijs (NVMO); reference number 924, 877, 653 and 2020.3.8).

### 2.4. Measurements

The questionnaire was made available in both English and Dutch.

#### 2.4.1. Personal Demands

We included three different personal demands: irrational performance demands, awfulizing, and irrational need for control. Items were retrieved and adapted from the work-related irrational beliefs questionnaire (WIB-Q) [23]. We used five items per personal demand and changed the work context of each item to match the student context of this study. Irrational performance demands capture the irrational beliefs students have regarding their performance and study goals. An example item is “I have to be the best at my studies”. Awfulizing refers to the irrational beliefs students have about failure and its consequences, such as “If I make a mistake in my studies, the consequences are terrible”. Finally, the irrational need for control captures the irrational beliefs students have about their need for control, and an example item is “I cannot stand having any ambiguity in my studies”. Participants were asked to rate each item on a scale ranging from 1 (completely disagree) to 5 (completely agree). Confirmatory Factor Analysis (CFA) revealed that a three-factor model had an acceptable fit to the data (ꭓ^2^ (87) = 403.82, *p* < 0.001, root mean square error of approximation (RMSEA) = 0.08, standardized root mean residual (SRMR) = 0.06, comparative fit index (CFI) = 0.92, Tucker–Lewis index (TLI) = 0.91). Furthermore, modification indices suggested correlating error terms between item 2 “I can only cope with study situations when they are predictable” and item 3 “I am able to cope with unexpected events in my studies”, which are both from the subscale irrational need for control. We therefore decided to exclude item 3, which increased the model fit substantially (ꭓ^2^ (74) = 293.90, *p* < 0.001, RMSEA = 0.07, SRMR = 0.05, CFI = 0.94, TLI = 0.93). Cronbach’s alphas for dysfunctional performance demands (α = 0.85), awfulizing (α = 0.86) and dysfunctional need for control (α = 0.83) were good.

#### 2.4.2. Study Demands

Study demands were operationalized with three indicators: emotional demands, workload, and study-home interference. Emotional demands (four items) and workload (five items) were measured with slightly adapted items from the measurement of the questionnaire on the experience and evaluation of work (i.e., the Dutch name is vragenlijst beleving en beoordeling van de arbeid) [31]. Emotional demands refer to the extent to which one experiences an emotional load from one’s studies. An example item for emotional demands is “Are you confronted with upsetting situations in your studies?”. Workload refers to the experienced amount of work and work pressure (e.g., “Do you have too much work to do?”). Study-home interference was measured with four items adapted from the Survey Work-home Interaction Nijmegen (SWING) questionnaire [32] and concerns the degree to which the respondent experiences spillover from the study into one’s private life and vice versa. An example item is: “How often does it happen that you have to cancel appointments with your spouse/family/friends due to study-related commitments?”. Emotional demands were measured with an answering scale ranging from 1 (never) to 4 (always), whereas workload and study-home interference were both measured using an answering scale ranging from 1 (never) to 5 (always). CFA revealed that the hypothesized three-factor structure fitted the data well (ꭓ^2^ (62) = 256.79, *p* < 0.001, RMSEA = 0.08, SRMR = 0.07, CFI = 0.94, TLI = 0.92). Reliability analysis revealed good internal consistencies for emotional demands (α = 0.83), workload (α = 0.82) and study-home interference (α = 0.85).

#### 2.4.3. Burnout

Burnout was measured using the 13-item version of the Burnout Assessment Tools (BAT) [33], which is available in English and Dutch. The BAT consists of four indicators. Exhaustion was measured with four items (α = 0.83), such as “When I’m studying, I feel mentally exhausted”. Mental distancing was measured with three items (α = 0.83). An example item of mental distancing is “I feel a strong aversion towards my studies”. Impaired emotional control is measured with three items (α = 0.88) and includes items such as “I may overreact unintentionally”. Impaired cognitive control is measured with three items as well (α = 0.88). An example item of impaired cognitive control is “When I’m studying, I have trouble staying focused”. Participants were asked to rate each statement on a scale ranging from 1 (never) to 5 (always). In order to adapt the scale to a student’s situation we decided to replace one (exhaustion) item (‘At work, I feel physically exhausted’) with another item from the full BAT instrument (‘When I exert myself in my studies, I get tired quicker than normal’). CFA confirmed that a four-factor model fitted the data well (ꭓ^2^ (59) = 263.27, *p* < 0.001, RMSEA = 0.08, SRMR = 0.05, CFI = 0.95 TLI = 0.94). Cronbach’s alpha for the total scale was excellent (α = 0.92).

### 2.5. Statistical Analyses

We analyzed the data using Mplus 7 [34]. Mplus uses the full information maximum likelihood (FIML) method to deal with missing data, which is recommended in social and behavioral research [35]. To test our hypotheses, we conducted a set of latent analyses on the basis of three models. In Model 1, we tested the associations between personal demands and burnout and between study demands and burnout (main effects model). In Model 2, we added the association between personal demands and study demands and tested the indirect relationship between personal demands and burnout via study demands (mediation model). In Model 3, we added the latent interaction effect between personal demands and study demands on burnout (moderation model). In order to model latent interactions in Mplus, we specified a random model that delivers a relative instead of an absolute fit measure [34]. Hence, we made use of the Akaike information criterion (AIC) and the Bayesian information criterion (BIC) to compare the fit measures of all three models. In general, a lower AIC and BIC were the decision criterion in favor of accepting a specific model [36].

## 3. Results

The descriptive statistics and correlation matrix between the study variables are shown in Table 1. As can be seen in Table 1, all correlations between study demands, personal demands, and burnout appeared to be positive.

### 3.1. Measurement Model

Multilevel confirmatory factor analysis was conducted to examine the validity of our measurement model. The proposed model included three latent variables (i.e., personal demands, study demands, and burnout). Results showed a better fit to the data for a model comprising the three latent factors (ꭓ^2^ (156.567) = 32, CFI = 0.94, TLI = 0.91, RMSEA = 0.08), compared with models that comprised two latent factors (model in which burnout and personal demands were merged ꭓ^2^ (375.995) = 34, CFI = 0.83, TLI = 0.77, RMSEA = 0.13; model in which burnout and study demands were merged ꭓ^2^ (257.782) = 34, CFI = 0.89, TLI = 0.85, RMSEA = 0.11; model in which personal and study demands were merged ꭓ^2^ (366.984) = 34, CFI = 0.83, TLI = 78, RMSEA = 0.13).

### 3.2. Hypothesis Testing

Table 2 presents fit indices for the main effects model (Model 1), the mediation model (Model 2) and the moderation model (Model 3). Absolute fit measures are provided for Model 1 (ꭓ^2^ (32) = 156.567, *p* < 0.001, RMSEA = 0.083, SRMR = 0.056, CFI = 0.94, TLI = 0.91) and Model 2 (ꭓ^2^ (33) = 160.309, *p* < 0.001, RMSEA = 0.082, SRMR = 0.056, CFI = 0.94, TLI = 0.91). Based on these indicators we could conclude that the main effects model (Model 1) and the mediation model (Model 2) both fitted the data well. As Mplus does not generate absolute fit measures for latent interactions in random models [33], there were no absolute fit measures available for the interaction model (Model 3).

Table 2 also presents the AIC and BIC for the three tested models. As can be seen in Table 2, the BIC value for the mediation model (Model 2) represents a slightly better fit to the data compared with the moderation model (Model 3; ΔBIC Model 3 compared to Model 2 = 2.822). In contrast, the AIC values suggest that the moderation model (Model 3) represents a slightly better fit to the data compared with the mediation model (Model 2; ΔAIC Model 3 compared to Model 2 = −1.52). According to the rule of thumb that Burnham and Anderson propose (2004), the AIC and BIC must differ at least 10 for one model to be better fitting than the other. A priori, we decided that we would continue to examine the parameters from each model when the fit measures would not differ meaningfully from each other. Hence, as the present fit measures differ less than 10, we had to conclude that both models fit the data equally well, and we continue with inspecting the parameters of the hypothesized relationships within each of the models.

To test Hypothesis 1, which predicted that personal demands would be positively associated with burnout via the perception of study demands, we examined the results of Model 2 (see Table 2). The results of Model 2 reveal a positive association between personal demands and burnout (*β* = 0.45, *b* = 0.60, *SE* = 0.05, *t* = 11.572, *p* < 0.001) as well as a positive relationship between study demands and burnout (*β* = 0.41, *b* = 0.60, SE = 0.05, *t* = 11.572, *p* < 0.001). This suggests that students experience more burnout symptoms with increasing levels of personal demands and study demands. Furthermore, the results from Model 2 indicate an indirect relationship between personal demands, via study demands, on burnout (*β* = 0.19, *b* = 0.26, *SE* = 0.05, *t* = 5.29, *p* < 0.001). The data thus provided supported Hypothesis 1. This means that with increasing personal demands, students perceive more study demands in their study, which in turn related positively to burnout. As the direct association between personal demands and burnout remains significant, it can be concluded that study demands partly mediated the positive relationship between personal demands and burnout.

Hypothesis 2 posited that personal demands would moderate the relationship between study demands and burnout. The results of Model 3 (Table 2) show that the interaction term of study demands and personal demands is not significant (*b* = −0.20, *SE* = 0.11, *t* = −1.76, *p* = 0.079). Hence, the relationship between study demands and burnout was not moderated by students’ personal demands, and we rejected Hypothesis 2.

## 4. Discussion

This study explored the role of personal demands in the health impairment process of the JD-R model. In order to do so, we first refined the conceptualization of personal demands to create a better fit with the JD-R model [9]. We subsequently conducted a survey study among master students in four faculties at a Dutch University. Based on the Transactional Model of Stress [17], we expected that personal demands could affect the health impairment process in two ways. By influencing the perception of the study situation to be more demanding (primary appraisal), personal demands may be associated with elevated levels of student burnout (i.e., mediation hypothesis). Furthermore, we theorized that personal demands might hinder students to successfully handle challenging study requirements and feel confident about their capability for successful enactment (secondary appraisal), which would increase the strength of the relationship between study demands and student burnout (i.e., moderation hypothesis).

Findings provided only support for the mediation hypothesis. Hence, with increasing levels of personal demands, students perceived more study demands, which in turn related positively with burnout symptoms. The results did not support the moderation hypothesis.

### 4.1. Theoretical Conceptualization of Personal Demands

Building upon a previous definition [20], we proposed a refined conceptualization of personal demands. Specifically, we decided to do so in order to align the definition of personal demands with the conceptualization of personal resources while providing a clear distinction between both constructs. First, we included a broader array of self-related aspects that may cause individuals’ energy to be consumed and not replenished, thereby streamlining our definition with the conceptualization of personal resources in the JD-R model [11]. Secondly, we aimed at stressing the maladaptive nature of personal demands by emphasizing that personal effort investment is disproportionate with regard to the outcomes that are pursued. Finally, we paid attention to the personal aspects that withhold individuals from properly responding to demanding aspects in their environment by including those personal aspects that restrict individuals to successfully cope with their environment. These adjustments resulted in the conceptualization of personal demands as “all aspects of the self that force individuals to invest disproportionate effort in their work and/or hamper them to successfully cope with their environment and are therefore associated with psychological and/or physical costs”.

### 4.2. Theoretical Contributions and Implications for Future Research

In line with the primary appraisal process of the Transactional Model of Stress [17], the present findings revealed that students with high personal demands tend to experience higher study demands. Hence, individuals who have a tendency to set irrationally high-performance standards, a high need for control, and a high tendency to awfulize are more inclined to appraise study situations as demanding. Moreover, the findings show that personal demands relate positively and indirectly to student burnout by influencing the perception of the study demands. These findings are in line with previous findings of a study showing that compulsive working is associated with perceiving more job demands and, indirectly, with higher levels of burnout [8]. Additionally, our findings underscore the theorizing of Bottos and Dewey [26], who suggested that perfectionistic students are more likely to generate their own stress because they are more likely to perceive situations as a hassle. Finally, our findings fit into a long line of studies showing that with increasing levels of study demands, students report more burnout symptoms [4,5,6,7,19].

Taken together, whereas only a few studies have examined the construct of personal demands [20], the results of the present and previous studies [8] suggest that personal demands can be considered as a predictor of the perceived level of study (or job) demands, which indirectly increases the risk on burnout. In other words, according to the present findings, personal demands may be regarded as a lens through which individuals perceive the demands in their study (or work) environment. These results point towards the relevance of including personal demands as a factor for understanding and investigating the occurrence and etiology of burnout in the context of a study (or work) environment.

Relating our results to the JD-R model, personal demands can be considered as a predictor of perceived study demands and work-related ill-being (i.e., burnout) in the health impairment process. Consequently, personal demands appear to have a similar place within the health impairment process as personal resources have within the motivational process [9]. Based on a variety of empirical studies (e.g., [11,37]), personal resources have been positioned as predictors of job resources and work-related well-being (i.e., work engagement) in the motivational process [9]. Hence, it appears that both personal resources and personal demands can be regarded as predictors of study (or job) resources and study (or job) demands, respectively, within the JD-R model.

At the same time, previous research revealed that personal resources might buffer the impact of job demands on strain [9,38]. The present study did not find a significant interaction effect between personal demands and study demands on burnout. Therefore, we concluded that our data do not lend support for a moderating role of personal demands in the health impairment process of the JD-R model. Nonetheless, because the present study is one of the first studies exploring the role of personal demands in the health impairment process, it is possible that other personal demands would alter the relationship between study demands and work-related ill-being (see [9]). Earlier studies do report that having an external locus of control, which can be viewed as a personal demand, relates to elevated levels of stress when individuals are confronted with demanding situations [28,29]. It would be interesting to see if current studies can replicate this finding with similar control-related personal demands, such as external locus of control or irrational need for control, as well as other personal demands, such as irrational beliefs or dysfunctional coping mechanisms. Furthermore, some personal resources have been found to boost the impact of job demands on motivation and, so to say, cross moderate between the processes within the JD-R model [13]. In a similar manner, personal demands could also cross moderate the motivational process within the JD-R model.

All in all, with the present findings, we offer initial information on the potential role of personal demands in the health impairment process of the JD-R model. Specifically, we demonstrate that irrational performance demands, awfulizing, and irrational need for control, as a latent construct of personal demands, predict the perceived level of study demands, which indirectly relates to burnout. Moreover, personal demands do not appear to moderate the association between study demands and burnout. We would advise future researchers to examine these and other personal demands within different contexts (e.g., employee contexts), using multiple designs (i.e., daily or weekly designs), and testing mediating, moderating, and cross moderating effects within the health impairment and the motivational process of the JD-R model.

### 4.3. Study Limitations

This study has some limitations that need to be addressed. First of all, the use of a cross-sectional design to test our hypotheses limits us in drawing causal conclusions and increases the risk of common method bias, for instance, by inflating the relationships among the study variables [39]. In order to test whether personal demands indeed influence the perception of study demands and burnout, interventions and longitudinal designs would be needed. However, as mentioned by Spector [40] (p. 129), it also “makes sense to start new areas of inquiry with the most efficient methods to provide initial evidence that a research question is deserving of attention”. Consequently, as our main aim was to operationalize the concept of personal demands and to explore its role in the JD-R model, we deemed it suitable to start with cross-sectional data to explore the relationships between personal demands, study demands, and burnout [9]. Another limitation of this study is that we subjectively measured study demands [31,32]. Although we selected validated concepts based on group interviews with students from all four faculties, we are unable to relate objective study demands to personal demands, subjective measurements of study demands, and student burnout. Future researchers may want to include objective measures of study demands, such as average study time or teacher ratings of the workload, to further unravel students’ appraisal processes in relation to their personal demands. Furthermore, since only 30% of the total invited students participated in the study, we should be cautious with generalizing the current findings to all graduate students. Finally, the generalizability to a broader student context (e.g., undergraduate students, students of applied universities, or high school students) may be limited as we measured a specific set of study demands and personal demands based on focus groups of graduates (i.e., master level) university students. Similarly, we should also be careful with generalizing the current findings towards student populations in other countries with varying cultures and school systems. Although we are confident that the chosen demands are relevant to the present student sample, in order to be able to generalize the present findings to a broader student population, it is important to replicate this study with different student samples from different countries.

### 4.4. Practical Implications

The present study reveals that personal demands may form a burden for students, as it relates to an unfavorable perception of their study environment. If this process endures for a considerable while, it may increase students’ chances of burnout. Personal demands may force students to invest disproportionate amounts of time and energy into their studies and may hinder energy replenishment. A practical implication for student counselors would be to assess relevant personal demands among students with increased risk of burnout and to raise awareness about the potentially harmful effects of personal demands in order to prevent burnout. Counselors may also try to reduce personal demands by actively challenging dysfunctional beliefs and underlying assumptions, using techniques that are commonly used in Cognitive Behavioural Therapy (cognitive restructuring) [41]. As the present study reveals that irrational performance demands, awfulizing, and an irrational need for control are relevant personal demands that may trigger health impairment, practitioners may, in particular, want to target these three demands during counseling. Additionally, student counselors may try to reduce the risk of burnout by practicing self-efficacy and coping skills [42].

## 5. Conclusions

The present study offers an elaborated and refined conceptualization of the construct of personal demands. In addition, we have uncovered the role of personal demands in the health impairment process of the JD-R model. Specifically, the study findings indicate that personal demands predict the perceived level of study demands, and indirectly, related to student burnout. Moreover, the present study shows that personal demands do not moderate the association between study demands and burnout. All in all, whereas a demanding study in itself appears to constitute a risk factor for students in developing burnout, the present study suggests that students’ own personal demands are a relevant factor to consider in both research and counseling settings.

## Figures and Tables

**Figure 1 ijerph-18-00632-f001:**
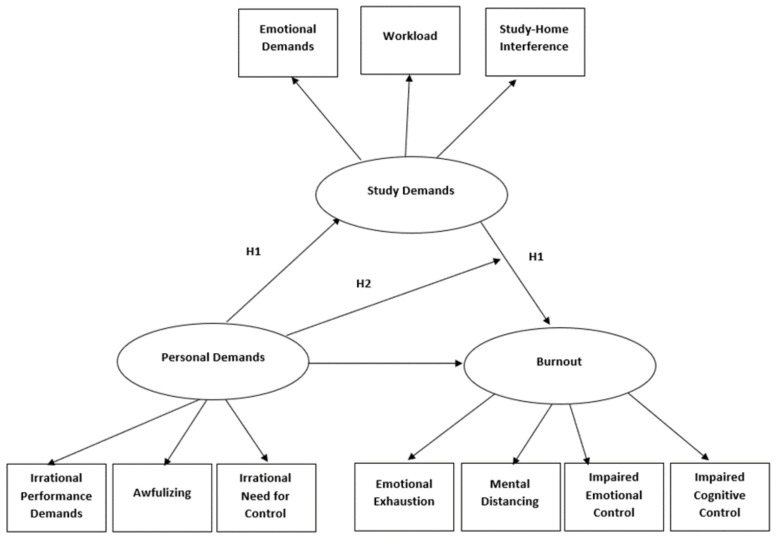
The research model depicting all indicators of the latent constructs.

**Table 1 ijerph-18-00632-t001:** Descriptive statistics and Pearson correlations for all study variables (*n* = 568).

	*M*	*SD*	1	2	3	4	5	6	7	8	9
Burnout											
1. Exhaustion	3.30	0.84	-								
2. Impaired Emotional Control	2.47	0.94	0.64 **	-							
3. Impaired Cognitive Control	2.75	0.87	0.61 **	0.54 **	-						
4. Mental Distancing	2.12	0.82	0.52 **	0.46 **	0.49 **	-					
Study Demands											
5. Work Pressure	3.07	0.80	0.37 **	0.22 **	0.19 **	0.11 **	-				
6. Study-Home Interference	2.84	0.86	0.53 **	0.42 **	0.35 **	0.24 **	0.52 **	-			
7. Emotional Demands	1.90	0.57	0.33 **	0.30 **	0.21 **	0.15 **	0.23 **	0.34 **	-		
Personal Demands											
8. Dysfunctional Performance Demands	3.26	0.83	0.28 **	0.30 **	0.15 **	0.05	0.22 **	0.21 **	0.04	-	
9. Awfulizing	3.11	0.84	0.45 **	0.44 **	0.34 **	0.26 **	0.33 **	0.34 **	0.19 **	0.61 **	-
10. Dysfunctional Need for Control	2.83	0.84	0.45 **	0.45 **	0.36 **	0.27 **	0.17 **	0.25 **	0.18 **	0.42 **	0.53 **

** *p* < 0.01. M: Mean; SD: standard deviation.

**Table 2 ijerph-18-00632-t002:** Regression coefficients and fit indices for the main effects model, the mediation model, and the moderation model with burnout as dependent variable (*n* = 568).

	Model 1 Main Effects				Model 2 Mediation/Indirect Relationships				Model 3 Moderation/Interaction		
Variables	*b (SE)*	*B*	*t*	*p*	*b (SE)*	*β*	*t*	*p*	*b (SE)*	*t*	*p*
Personal Demands	0.48 (0.09)	0.36	5.246	<0.01	0.60 (0.05)	0.45	11.572	<0.01	0.48 (0.09)	5.571	<0.01
Study Demands	0.75 (0.12)	0.49	6.441	<0.01	0.60 (0.05)	0.41	11.572	<0.01	0.77 (0.11)	6.781	<0.01
Personal Demands → Study Demands					0.26 (0.05)	0.19	5.289	<0.01			
Personal Demands × Study Demands									−0.20 (0.11)	−1.756	0.079
RMSEA	0.083				0.082				-		
SRMR	0.056				0.056				-		
CFI	0.94				0.94				-		
TLI	0.91				0.91				-		
AIC	11,620.481				11,622.223				11618.961		
∆ AIC	-				∆ 1.742				∆ −1.52		
BIC	11,763.771				11,761.171				11,766.593		
∆ BIC	-				∆ −2.6				∆ 2.822		
Degrees of freedom	33				32				34		

Note. Unstandardized coefficients (*b*) and standardized coefficients (*β*) are reported for Model 1 and Model 2. For Model 3, only unstandardized coefficients (*b*) are reported since Mplus does not provide standardized coefficients for the random latent interaction model. Furthermore, only relative fit measures were available for Model 3. S.E: Standard Error; *t* = t-value; *p* = *p*-value RMSEA: root mean square error of approximation; SRMR: standardized root mean residual; CFI: comparative fit index; TLI: Tucker–Lewis index; AIC: Akaike information criterion; BIC Bayesian information criterion.

## Data Availability

The data presented in this study are available on request from the corresponding author. The data are not publicly available due to privacy issues and the continuing of the data collection for a longitudinal project.

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
