# Peer review of "Exploring the Role of Personal Demands in the Health-Impairment Process of the Job Demands-Resources Model: A Study among Master Students"

_ijerph, 2021, doi:10.3390/ijerph18020632_

Round 1

Reviewer 1 Report

This is an important and interesting study. Suggesting some minor revisions and clarifications before publication.

Line 34-35 - citations are confusing. (e.g., 8) doesn’t make sense. Would suggest a clarification.

Line 39: what is an example of an ‘emotional hassle’? How are those difference than personal demands

Line 50: I would be interested to see a sentence or two that explains briefly what these three exception studies examined.

Line 104: Referencing needs to be looked at and the sentence restructured.

Line 110: Replace ‘Taking’ with ‘Taken’.

Line 156: check reference for [i.e., 26] change to provide the “as in” rather than the reference number.

Overall comments:

-authors refer to aa 'disproportionate amount of effort' but do not provide a comparator. What is the effort disproportionate to? Compared to what outcome measurement? (Grades? Satisfaction with performance?) How is that measured? How is it measured - By the individual? Or the institution? And how would that impact the individual?

-Are the differences between the three study programs similar enough to be lumped together in the analysis? Was heterogeneity of the three samples tested?

Lines 194-196: Were these the sample size targets? If these are the actual sample sizes, would move to the results section. Here in the methods, can you discuss the reason you picked 2 vet groups, and why each sample size was selected? How were students recruited for the focus groups? Were they provided with an honorarium? Who conducted the focus groups and what was their relationship to the students?

Was the content analysis conducted by one author or multiple analysts? What methods of validation for qualitative analysis were used to ensure the rigour of the results?

When the questionnaire was developed, was it piloted by anyone? Brought back to the focus group participants to ensure it was representative of their experiences?

Lines 300-304. Suggest adding a sentence to the methods to indicate what the a-priori decision was around how to interpret a model when the AIC/BIC results are contradictory

Limitations

- response rate was not discussed

-talk about the qualitative rigour to help justify the subjective study demands

- what is the possible impact of a "common method bias" in the context of this study?

Reviewer 2 Report

This paper presents a study on the role of personal demands for the well-being of master students from four different faculties of one university in the Netherlands. For this purpose, they used the Job Demands-Resources (JD-R) Model.

Suggestions and questions (answers can/should be used to improve the paper):
1. The name 'Job Demands-Resources' could be presented in the title - not the abbreviation JD-R.
2. Sections and subsections should be numbered.
3. Demographic information of the participants was not presented. At least age should be presented.
4. It is a little bit weird the way that section 3 is presented: a specific section (3.2. Figures, Tables and Schemes) only to present tables and figures. That content (tables e figures) would flow better if presented along with the text.
5. How generalized can the results be? Discussion considered different 'types' of students, but could other aspects be considered? For example, age, country, culture, duration of the course (total time and current time), etc. A deeper discussion on this would motivate future research.

Specific comments:
- lines 34-35: "[e.g., 8] (e.g., Guglielmi et al., 2012)" - two reference formats?
- line 75: "[20] (p. 751)" - idem (there are more occurrences)
- line 189: Divide the paragraph starting on this line. It presents different contents.
- Paragraphs in lines 213, 231, and 247 could be organized as (sub)subsections.
- "4.1. Theoretical Conceptualization Personal Demands" -> 4.1. Theoretical Conceptualization OF? Personal Demands
